# APRIL: ACTIVE PARTIAL ROLLOUTS IN REINFORCEMENT LEARNING TO TAME LONG-TAIL GENERATION

## ABSTRACT

Reinforcement learning (RL) has become a cornerstone in advancing large-scale pre-trained language models (LLMs). Successive generations—including GPT-o series, DeepSeek-R1, Kimi-K1.5, Grok 4, and GLM-4.5—have relied on large-scale RL training to enhance reasoning and coding capabilities. To meet the community's growing RL needs, numerous RL frameworks have been proposed. Most of these frameworks primarily rely on inference engines for rollout generation and training engines for policy updates. However, RL training remains computationally expensive, with rollout generation accounting for more than 90% of total runtime. Besides, its efficiency is often constrained by the long-tail distribution of rollout response lengths, where a few lengthy responses stall entire batches, leaving GPUs idle and underutilized. As model and rollout sizes continue to grow, this bottleneck increasingly limits scalability. To address this challenge, we propose Active Partial Rollouts in Reinforcement Learning (APRIL), which mitigates long-tail inefficiency. In the rollout phase, APRIL over-provisions rollout requests, terminates once the target number of responses is reached, and recycles incomplete responses for continuation in future steps. This strategy ensures that no rollouts are discarded while substantially reducing GPU idle time. Experiments show that APRIL improves rollout throughput by **22.5**% on average across commonly used RL algorithms (GRPO, DAPO, GSPO), accelerates convergence, and achieves **2.1**% higher final accuracy across tasks on average. Moreover, APRIL is both framework- and hardware-agnostic, already integrated into the slime[1] RL framework, and deployable on NVIDIA and AMD GPUs. Taken together, this work unifies system-level and algorithmic considerations in proposing APRIL, with the aim of advancing RL efficiency and inspiring further optimizations in RL systems.

## 1 INTRODUCTION

Reinforcement learning (RL) has become a critical phase in the development of large-scale pre-trained language models (LLMs) (OpenAI, 2023; DeepSeek-AI et al., 2025; Kimi et al., 2025; OpenAI et al., 2025). Its importance was first highlighted by the release of ChatGPT (OpenAI et al., 2025) at the end of 2022, when reinforcement learning from human feedback (RLHF) (Ouyang et al., 2022) proved highly effective in aligning LLMs with human instructions. RL-trained LLMs such as ChatGPT and Claude subsequently demonstrated substantially improved abilities to follow human instructions. Inspired by this, recent LLMs, including the GPT-o series (OpenAI et al., 2025), DeepSeek-R1 (DeepSeek-AI et al., 2025), Kimi-K1.5 (Kimi et al., 2025), and GLM-4.5, have employed large-scale RL training to enhance their capabilities, particularly in reasoning and coding. However, as RL continues to drive rapid progress in LLM capabilities, the associated computational and efficiency challenges have grown correspondingly. This underscores the urgent need for more scalable and efficient RL training frameworks to fully realize the potential of next-generation LLMs.

This need has been actively addressed by the open-source community, resulting in scalable RL training frameworks such as OpenRLHF (Hu et al., 2025), verl (Sheng et al., 2025), Areal (Fu et al., 2025), and slime (THUDM, 2025). These frameworks typically integrate advanced distributed engines to enhance efficiency—employing vLLM (Kwon et al., 2023) or SGLang (Zheng et al., 2024) as inference backends for LLM rollouts, FSDP (Zhao et al., 2023) or Megatron-LM (Shoeybi et al.,

---

[1]slime repository: `https://github.com/THUDM/slime`

2019) as training backends for LLM optimization, and Ray (Moritz et al., 2018) for orchestrating parallel training and inference to maximize resource utilization during RL training.

Despite extensive infrastructure-level optimizations, the rollout phase remains the dominant bottleneck: each input requires the LLM to generate $\mathcal{N}$ responses of length up to $\mathcal{L}$ in an auto-regressive manner, which **typically accounts for over 90% of the total RL training runtime** (Hu et al., 2025). Besides, the response length of the rollouts for each input instance often varies significantly in most tasks (3.2). As a result, in any given training batch, a small number of examples may require exceptionally long responses. This "long-tail" distribution **causes the entire batch to stall until the longest-running instance rollout is completed**, leading to severe inefficiency and under-utilization of computational resources, as faster-generating workers remain idle (Zhong et al., 2025c). Moreover, with recent findings and the emerging trend of scaling up rollout inference ($\mathcal{N}$ and $\mathcal{L}$) to enhance LLM capabilities (Snell et al., 2024; Muennighoff et al., 2025), the increasing time and computational cost on the rollout side are becoming a fundamental performance bottleneck in scaling on-policy RL training (Schulman et al., 2017; Rafailov et al., 2023; Mroueh et al., 2025; Yu et al., 2025b).

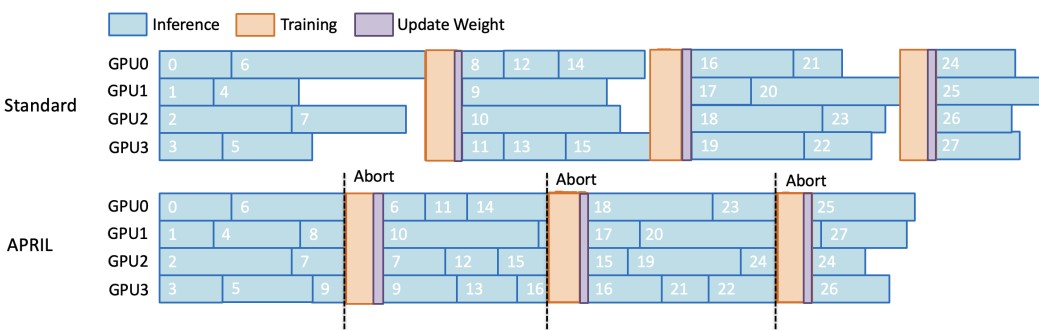

Figure 1: (Above) In the standard synchronous RL training paradigm, GPU utilization is often suboptimal. (Below) Our APRIL mechanism mitigates this issue by reducing the bubble during RL.

Thereby, this raises a critical question: *How can we optimize rollouts to reduce inference overhead, especially under long-tail rollout generation in RL training?* To address this challenge, we propose *Active Partial Rollouts* (APRIL), a compute-efficient method to accelerate rollout generation in RL training. In each iteration, we deliberately over-provision rollout requests (exceeding the default batch size) to the inference engines. Once the default number of rollouts is reached, the engines actively stop the remaining unfinished rollouts. Instead of discarding these partial results, we store them in a buffer and resume their generation in the next iteration. Meanwhile, the completed rollouts are used to compute the loss and optimize the LLM. This approach not only safeguards model accuracy by ensuring that no rollout instances are discarded but also systematically recycles incomplete rollouts to alleviate the long-tail effect, thereby substantially reducing GPU idle time during inference. As a result, throughout the RL training process, APRIL markedly enhances overall training efficiency while preserving the original accuracy. In practice, our experiments demonstrate that APRIL not only delivers a significant improvement (3.2.1) in rollout throughput (up to **49.5**% at most) across widely used RL algorithms—GRPO (Shao et al., 2024), DAPO (Yu et al., 2025b), and GSPO (Zheng et al., 2025)—as well as various LLMs (DeepSeek-AI et al., 2025; Team, 2025), but also achieves faster convergence and higher final accuracy (3.2.2). Moreover, we make additional efforts in GPU memory management to support deployment across hardware platforms, such as NVIDIA and AMD (3.3).

In summary, our contributions are:

- We **quantitatively characterize the long-tail generation problem**, demonstrating that rollout lengths vary significantly within the same batch and thereby underscoring the necessity of improved rollout strategies in RL to improve efficiency without compromising accuracy (3.2).
- We are **the first to propose and open-source Active Partial Rollout (APRIL)**, which **improves rollout throughput by 22.5**% on average across widely algorithms in RL training (3.2.1).
- In practice, our experiments demonstrate that APRIL not only improves the RL efficiency, but **achieves faster convergence and increases the final accuracy by around 2.1**% (3.2.2).
- APRIL is readily deployable across RL frameworks, has already been integrated into slime (THUDM, 2025), and is compatible with both NVIDIA and AMD GPU platforms (3.3).

## 2 METHOD

### 2.1 PRELIMINARY ABOUT GENERAL RL TRAINING

Before introducing APRIL, we first formalize the standard reinforcement learning (RL) training process. For clarity, we adopt the REINFORCE algorithm (Williams, 1992) as a general formulation of RL training (Yao et al., 2025), which can be expressed as:

$$\theta_{k+1} \leftarrow \theta_k + \mu \cdot \underbrace{\mathbb{E}_{a \sim \pi(\theta_k)}}_{\text{Inference: rollout}} \Big[ R(a) \cdot \underbrace{\nabla_\theta \log \pi(a, \theta_k)\big|_{\theta=\theta_k}}_{\text{Training}} \Big], \tag{1}$$

where $\mu$ denotes the learning rate, and $\theta_k$ represents the policy model parameters at step $k$. The term $a \sim \pi(\theta_k)$ indicates that the action $a$ is sampled from the policy distribution defined by the current parameters $\theta_k$, which corresponds to the *rollout* process in the inference engine. The gradient term $\nabla_\theta \log \pi(a, \theta_k)\big|_{\theta=\theta_k}$ is the score function, measuring how the log-probability of the sampled action changes with respect to the policy parameters. This component can be viewed as the *training* step, as it determines how to adjust the policy parameters $\theta$ to increase the expected reward $R(a)$. It's worth noting that $R(.)$ could be a learned reward model, as in PPO (Schulman et al., 2017), or a reward function, as in GRPO (Mroueh et al., 2025), DAPO (Yu et al., 2025a), and GSPO (Zheng et al., 2025).

### 2.2 PARTIAL ROLLOUT FOR MITIGATING THE LONG-TAIL PROBLEM

In the standard synchronous RL training paradigm, given a batch of instances, the inference engines generate rollouts for all instances ($a \sim \pi(\theta_k)$) until completion. The resulting trajectories are then forwarded to the training engine ($\nabla_\theta \log \pi(a, \theta_k)\big|_{\theta=\theta_k}$) to update the policy model, as illustrated in Formula 1 and Figure 2. However, this procedure often leads to suboptimal GPU utilization, since faster rollouts must wait for the longest sequences to finish in the inference engine, thereby introducing idle time and slowing down the overall training cycle.

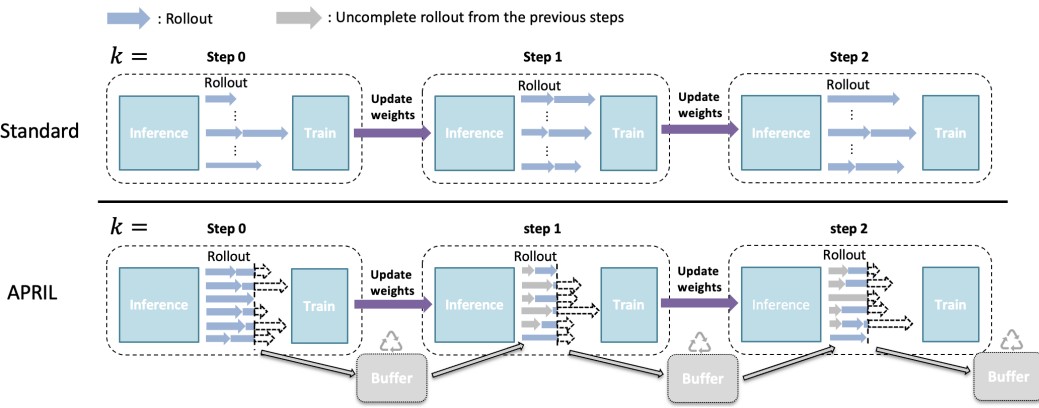

Figure 2: APRIL over-provisions instances, stops once the target number of instances finishes their rollouts, and updates the policy model. Incomplete rollouts are stored in the data buffer and resumed in later steps, thereby reducing idle time and improving training efficiency.

**APRIL: Active Partial Rollouts in Reinforcement Learning.** To mitigate inefficiency, we propose the APRIL mechanism as shown in Figure 2. Instead of waiting for all workers to finish rollouts in the inference stage, APRIL launches an over-provisioned number of instance rollouts ($N' > N$) and terminates once the required $N$ instance rollout sequences are completed at step $k$. The finished rollouts are used to optimize the policy model $\pi(\theta_k)$, while the unfinished ones are buffered and resumed in the next step $k + 1$, reducing GPU idle time without data loss. The process is as follows:

- **Over-provisioned Generation:** Start $N'$ instances where $N' > N$.
- **Early Termination:** Stop generation once $N$ instance rollout sequences are completed.
- **Buffering Continuations:** Store unfinished instance rollout rollouts in a continuation buffer; completed ones are sent for policy optimization.

- **Prioritized Resumption:** At step $k + 1$, resume buffered rollouts before starting new ones.

APRIL approach significantly improves system throughput by eliminating delays caused by the completion of "long-tail" rollout generations. However, an important point should be noted next.

**Implications for the Advantage Function.** APRIL mechanism introduces a subtle but important deviation from the strict on-policy assumption of RL algorithms. For instance, a training batch $N$ used to update the policy model parameters to $\theta_{k+1}$ is now composed of multiple distinct sets of data:

- **Newly-completed rollout instances ($\mathcal{D}_{k+1 \ (completed)}$):** Trajectories that were initiated and completed entirely under the current policy $\pi(\theta_{k+1})$.
- **Continued rollout instances ($\mathcal{D}_{k \ (completed)}$):** Trajectories was initialized by an earlier policy $\pi(\theta_k)$, then aborted and resumed one or more times, and ultimately completed under the current policy $\pi(\theta_{k+1})$.

Consequently, the collected rollout dataset $N$ in $\mathcal{D}_{k+1}$ is not monolithic. Instead, it comprises data generated not only from $\pi(\theta_{k+1})$ but also from preceding policies $\pi(\theta_k), \pi(\theta_{k-1}), \ldots, \pi(\theta_{k-m})$, where $m \leq k$.

Therefore, the advantage estimate associated with such trajectories is calculated over a sequence generated by a "mixed" policy. Our empirical results demonstrate that this modification does not destabilize training and occasionally yields modest but consistent gains, as shown in Figure 7. Our method has been successfully applied to multiple most common used RL algorithms, including PPO (Schulman et al., 2017), GRPO (Mroueh et al., 2025), DAPO (Yu et al., 2025a), and GSPO (Zheng et al., 2025).

## 3 EXPERIMENTS

The objective of our experiments is to validate the effectiveness of APRIL in mitigating long-tail rollout generation by: (1) analyzing the severity of long-tail issue in RL training, especially during rollouts (3.2); (2) evaluating throughput to verify APRIL's efficiency (3.2.1); (3) assessing convergence and accuracy to verify robustness (3.2.2); and (4) examining potential trade-offs (3.2.3).

### 3.1 EXPERIMENTAL SETUP

**Algorithms, Model, and Datasets** Our experiments are conducted on two models `Qwen3-4B`[2] (Yang et al., 2025) and `Qwen3-8B`[3] (Yang et al., 2025) with three math reasoning tasks. We use a diverse set of mathematical reasoning datasets for training, including `DAPO-Math-17k`[4], `DeepMath-103K`[5], and `DeepScaleR`[6]. To evaluate the final performance, we use the AIME-2024 benchmark[7], a collection of recent challenging math reasoning problems. This benchmark serves as a standard for assessing the advanced mathematical reasoning abilities of large language models. To faithfully validate the robustness of the proposed method, we apply APRIL to the two most widely used RL algorithms: GRPO (Mroueh et al., 2025) and DAPO (Yu et al., 2025a).

**Hyperparameter Setting and Hardware Platform** The hyperparameters are provided in Appendix A.2. We consider two configurations: the **standard setting (non-partial rollout)** and the **APRIL setting (partial rollout)**. For standard setting, `rollout_batch_size=32` denotes the number of input instances per training step. With `n_samples_per_prompt=8`, each instance produces multiple rollouts, yielding $32 \times 8 = 256$ samples per step. For APRIL setting, except for the above hypermeters, we over-provision rollouts by setting `over_sampling_batch_size=64` ($2\times$ `rollout_batch_size`), so that 512 rollouts are requested but the process terminates after the first 256 are completed. Besides, APRIL is designed to run efficiently on heterogeneous hardware. We conduct experiments primarily on a single node with $8\times$ NVIDIA H100 or $8\times$ AMD MI300 GPUs, but due to space constraints, the reported results focus on the AMD MI300 configuration.

---

[2]https://huggingface.co/Qwen/Qwen3-4B
[3]https://huggingface.co/Qwen/Qwen3-8B
[4]https://huggingface.co/datasets/open-r1/DAPO-Math-17k-Processed
[5]https://huggingface.co/datasets/zwhe99/DeepMath-103K
[6]https://huggingface.co/datasets/agentica-org/DeepScaleR-Preview-Dataset
[7]http://huggingface.co/datasets/HuggingFaceH4/aime_2024

## 3.2 THE LONG-TAIL PROBLEM IN ROLLOUT

To quantify the long-tail problem, we first conducted a pure rollout experiment without model training across three datasets. In our analysis, we selected representative datasets and generated four rollouts for each input instance, averaging their lengths per instance. All datasets exhibit long-tail behavior, particularly DeepMath-103K, for which the maximum generation length of 32,768 tokens was imposed to fully capture the long-tail distribution in analysis.

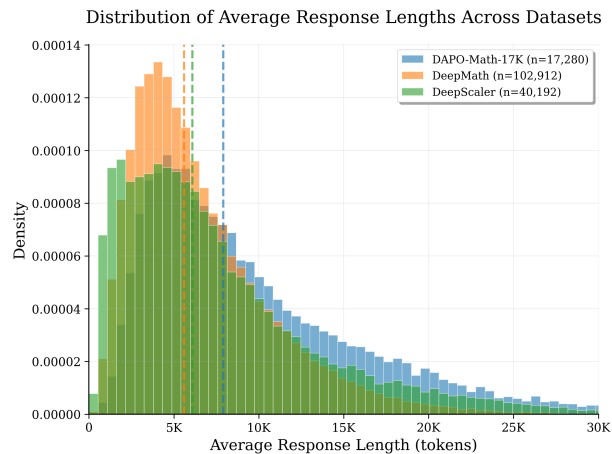

As shown in Figure 3, in all datasets , their rollout length distribution indicates that more than half of the rollout responses terminate within a few thousand tokens, while a long tail of outliers extends close to the maximum limit. This skewness implies that, in a standard synchronous RL framework, the completion time of a batch is dominated by the longest-generating instances. Consequently, substantial GPU idle time occurs during inference, leading to training bubbles, as illustrated in Figure 1 (Above).

Figure 3: Distribution of rollout response lengths reveals a pronounced long-tail peak in rollout of RL.

### 3.2.1 PERFORMANCE - TOKEN THROUGHPUT

A central claim of our work is that APRIL can substantially accelerate the rollout phase of the RL.

We present a comprehensive evaluation of our Active Partial Rollout (APRIL) method against a standard non-partial rollout baseline. Specifically, we measure *rollout throughput*, defined as *the total number of tokens generated on 8 GPUs divided by the wall-clock time per rollout iteration*. As shown in Table 1, we list the actual throughputs. For the Qwen3-4B model, APRIL consistently improves throughput by **24.4%**, **31.8%**, **37.7%** with GRPO algorithm, and **9.0%**, **13.5%**, **9.8%** with DAPO algorithm across three datasets. As for Qwen3-8B model, the throughputs improve by **26.4%**, **34.7%**, **49.5%** and **8.7%**, **8.5%**, **10.2%**, respectively[8]. Overall average throughput improves by around **22.5%**.

Table 1: Comparison of rollout throughput between the **baseline (non-partial rollout)** and **APRIL**. APRIL has the higher throughput across datasets: dapo-math-17k , DeepScaler, and DeepMath-103K.

| Alg. | Method | Qwen3-4B | | | | Qwen3-8B | | | |
| | | DAPO-Math-17k | DeepScaleR | DeepMath | Avg Imp. | DAPO-Math-17k | DeepScaleR | DeepMath | Avg Imp. |
|---|---|---|---|---|---|---|---|---|---|
| GRPO | APRIL | **12,210** | **12,183** | **13,027** | **+ 31.3%** | **11,096** | **10,821** | **11,594** | **+ 36.9%** |
| | Baseline | 9,814 | 9,252 | 9,459 | | 8,780 | 8,031 | 7,756 | |
| DAPO | APRIL | **13,698** | **13,916** | **14,318** | **+ 10.8%** | **12,407** | **12,439** | **13,075** | **+ 9.1%** |
| | Baseline | 12,572 | 12,257 | 13,038 | | 11,419 | 11,470 | 11,860 | |

### 3.2.2 CONVERGENCE SPEED AND ACCURACY

A potential concern with APRIL is that introducing off-policy rollouts (i.e., rollouts generated by earlier versions of the policy models , i.e., $\pi(\theta_k), \pi(\theta_{k-1}), \ldots, \pi(\theta_{k-m})$, where $m \leq k$, as described in Section 2.2 could destabilize training or degrade convergence and final accuracy in RL training.

To investigate this, we tracked the learning curves of models trained with both the baseline and APRIL methods, evaluating their performance on the AIME-2024 benchmark. As shown in Table 2, for the

---

[8]Refer to Appendix A.3.1 for a comprehensive throughput comparison across full training trajectories.

Table 2: We compare the accuracy of the **baseline (non-partial rollout)** and **APRIL**. Overall, APRIL performs comparably to the baseline and, in some cases, achieves slightly better final accuracy (**Acc.**).

| Alg. | Method | Qwen3-4B | | | | Qwen3-8B | | | |
| | | DAPO-Math-17k | DeepScaleR | DeepMath | Avg Imp. | DAPO-Math-17k | DeepScaleR | DeepMath | Avg Imp. |
|---|---|---|---|---|---|---|---|---|---|
| GRPO | APRIL | **65.8%** | **63.7%** | **71.2%** | + 3.3% | **66.9%** | 66.0% | **68.1%** | + 0.8% |
| | Baseline | 65.2% | 62.6% | 63.0% | | 66.5% | **66.7%** | 65.5% | |
| DAPO | APRIL | 68.2% | **67.8%** | **66.3%** | + 4.6% | **70.6%** | 68.9% | 71.4% | - 0.4% |
| | Baseline | **68.3%** | 66.6% | 53.5% | | 69.9% | **70.5%** | **71.8%** | |

Qwen3-4B model, APRIL improves accuracy by **0.6%**, **1.1%**, and **8.2%** on GRPO, and by **-0.1%**, **1.2%**, and **12.8%** on DAPO. For the Qwen3-8B model, APRIL achieves gains of **0.4%**, **-0.7%**, **2.6%** on GRPO, and **0.7%**, **-1.6%**, **-0.4%** on DAPO. Overall average accuracy improves by around **2.1%**.

Overall, APRIL does not degrade performance and, in some cases, achieves notable improvements. This suggests that incorporating mildly off-policy rollouts enhances rollout diversity, thereby positively influencing both learning dynamics and final model performance. Moreover, training remains stable and can even benefit from the added diversity when updating policies within the RL framework. We further observed that APRIL provides greater training stability[9]. In several baseline runs, we noted a phenomenon where response lengths abruptly exploded late in training, causing all samples to hit the `max_length` limit, which in turn led to drops in both rewards and evaluation scores. This issue was not observed in any of our APRIL runs, suggesting that the inclusion of slightly off-policy rollouts may act as a regularizer, preventing the policy from diverging into pathologically long generation modes. Thus, partial rollouts in APRIL may offer the additional benefit of improved training robustness. However, we did not encounter any cases where completing a rollout required more than five successive policy versions, i.e., $\pi(\theta_k), \pi(\theta_{k-1}), \ldots, \pi(\theta_{k-m})$, where $m \leq 5$. If rollouts were to depend on too many earlier policies, such excessive off-policy influence might lead to adverse effects. This remains an open question and a promising direction for future exploration.

### 3.2.3 ANALYSIS OF PERCENTAGE OF PARTIAL ROLLOUTS AND ROLLOUT LENGTH

From both performance and accuracy perspectives, APRIL demonstrates strong results. To further elucidate its effectiveness, we present a comprehensive analysis of its underlying mechanisms.

**How much off-policy data does APRIL introduce?**  A key feature of our partial rollout method is that tokens from incomplete trajectories are cached and reused in subsequent steps, as illustrated in Figure 2. This process inherently introduces off-policy data. We analyzed the rollout composition of our training batches and found this effect to be substantial. As shown in Figure 4, in a typical step with over-sampling, nearly all unfinished responses among the 256 additional requests are partially generated and then resumed in the next iteration. Tokens carried over from these truncated instances constitute approximately **40%** of the tokens in the subsequent steps. Despite this high proportion, APRIL remains remarkably stable

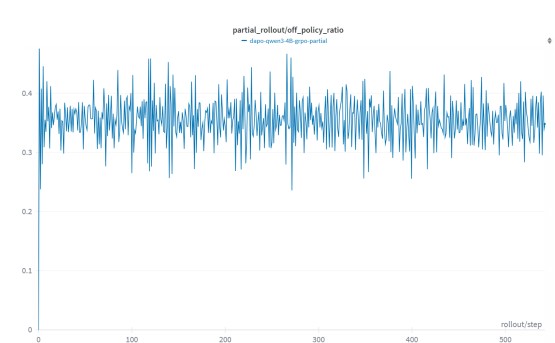

Figure 4: Our analysis indicates that approximately 40% of rollouts incorporate sequences generated by preceding policy models in each RL training step.

throughout training and even achieves slightly higher final accuracy. This observation suggests that incorporating rollouts from mutiple preceding policy models (within 5 steps as we aforementioned in Section 3.2.2 ) may diversify the rollout data combination and enhance the accuracy in RL training.

---

[9]Refer to Appendix A.3.2 for detailed accuracy results across training trajectories.

**Does instance-level group control reintroduce the long-tail phenomenon within intra-rollouts?**
In the previous APRIL framework, we focused on partial rollouts at the batch level, where rollout lengths exhibit substantial variability across input instances within a batch, as illustrated in Figure 3. It is important to note that when APRIL determines whether an input instance has completed its rollout, we adopt an *instance-level completion criterion*. Specifically, in our implementation, each input instance forms a group that encompasses all rollout responses associated with that instance. For example, when n_samples_per_prompt= 8, each input instance generates eight rollout sequences. The instance, together with its corresponding rollouts, is then collected and passed to the training engine to optimize the policy model. If an instance produces fewer than eight rollouts, the completed rollouts are retained in the data buffer until the remaining rollouts are generated, after which they are collectively used to compute the reward for updating the policy model.

A potential concern is that instance-level granularity might introduce an intra-group long-tail effect, where generated rollouts for a single instance might vary significantly in length. To examine this, we compare rollout-length variability at two levels of granularity: batch level and instance level.

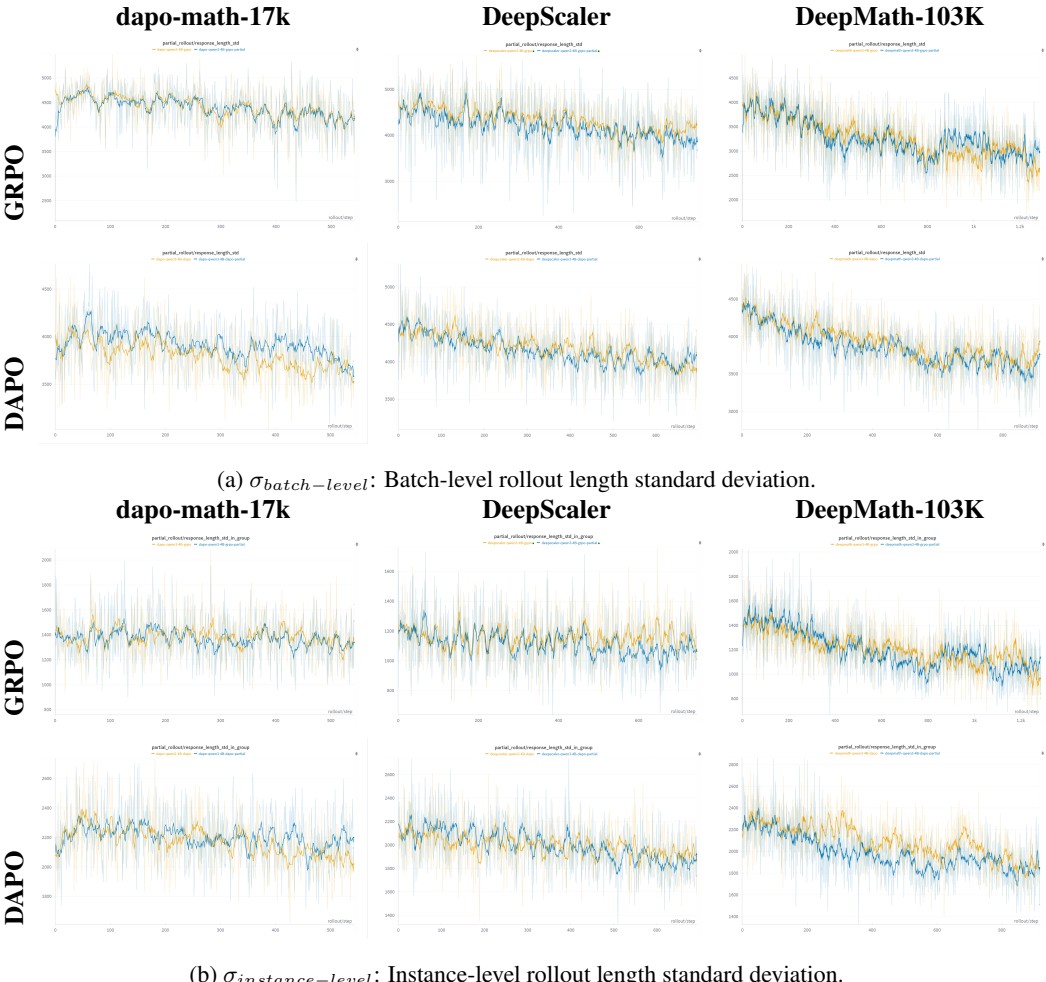

(a) $\sigma_{batch-level}$: Batch-level rollout length standard deviation.

(b) $\sigma_{instance-level}$: Instance-level rollout length standard deviation.

Figure 5: The x-axis represents the training steps, while the y-axis represents the standard deviation. The baseline (non-partial rollout) is in ——, and APRIL is in ——. We compare the standard deviation of response lengths per iteration at the batch level ($\sigma_{batch\text{-}level}$) and the instance level ($\sigma_{instance\text{-}level}$). The $\sigma_{instance\text{-}level}$ is apparently smaller than $\sigma_{batch\text{-}level}$; thus, there are fewer intra-group long-tail problems.

At the **batch level**, the standard deviation $\sigma_{batch-level}$ of response lengths across all prompts within an iteration is approximately 4,000-4,500 tokens across datasets as shown in Figure 5a. In contrast, at the **instance level** (within the same input instance), the $\sigma_{instance-level}$ is typically less than 1,500 tokens for GRPO algorithm, and 2,400 tokens for DAPO algorithm as shown in Figure 5b. This

pronounced gap indicates that responses sampled from the same input instance are considerably more homogeneous in length compared to responses sampled across different instances. Consequently, instance-level completion does not reintroduce a severe intra-group long-tail effect. Due to space limitations, we only present the Qwen-4B results, while more Qwen-8B results are in the appendix[10].

### 3.3 HARDWARE-AGNOSTIC

To demonstrate both the compatibility and general applicability of APRIL, we integrated it into the widely used open-source RL framework `slime` and replicated our main experiments across multiple hardware platforms, including NVIDIA H100 and AMD MI300. For deployment on AMD MI300 GPUs, we additionally implemented `torch_memory_saver` in HIP[11] on the AMD side. It is a PyTorch extension designed to alleviate GPU memory pressure during the alternating phases of inference and training under a co-allocated mode, where the training engine and inference engine share the same set of GPUs. This setup requires additional dispatching mechanisms in RL training.

`torch_memory_saver` introduces a region-based tagging mechanism that allows tensors (e.g., model weights, KV caches, and buffers) to be selectively managed. Specifically, tensors associated with a given tag can be *paused*, where their physical memory is released while their virtual addresses remain reserved. When needed, these tensors can be *resumed* by remapping newly allocated physical memory back to the original virtual addresses, thereby preserving CUDA/HIP Graph compatibility. In practice, this mechanism enables selective pausing and multi-stage resumption of memory regions, allowing users to release GPU memory after inference and restore it before subsequent rollouts. The key advantage is that it reduces out-of-memory errors and peak memory consumption without sacrificing performance or requiring modifications to CUDA/HIP Graph execution. With this, it can allow us to conduct RL training in co-allocated mode. The detailed implementation involves extensive discussion of low-level GPU operations and is therefore not emphasized here. The HIP-based implementation[12] has already been merged into the official upstream GitHub repository and has also been introduced in the online documents[13], which can be referenced in the footnote below.

## 4 RELATED WORK

In related work, we provide additional background and analyze the system-level design and algorithmic requirements in existing RL training to further clarify the motivation for creating APRIL.

**RL Framework Architectures: Synchronous vs. Asynchronous**   A classic mathematical formulation of RL is introduced in Equation 1. In practice, the mainstream implementation follows a synchronous RL paradigm, as exemplified by frameworks such as TRL (von Werra et al., 2020), OpenRLHF (Hu et al., 2025), verl (Sheng et al., 2025), and ROLL (Wang et al., 2025). In this paradigm, the inference and training engines operate as sequential processes. Specifically, when RL training is launched, a batch of instances is dispatched to the inference engine to generate rollouts. All workers simultaneously generate a full batch of experiences using a unified frozen policy model. Only after the entire batch has been collected does the policy update occur in the training engine. This design leverages the inherent on-policy nature of RL, ensuring that the training process consistently utilizes the most recent rollout data. These rollouts, directly sampled from the current policy model, promote training stability and facilitate convergence. However, this simplicity comes at a considerable cost to efficiency. In particular, the system must wait for all instances within a batch to complete rollout generation. Due to the long-tail phenomenon in rollout lengths (Section 3.2), some instances take substantially longer to finish, leading to severe underutilization of computational resources.

In response to the efficiency challenges inherent in synchronous RL, researchers and engineers have increasingly explored more dynamic and decoupled architectures. Fully asynchronous RL frameworks are emerging as a new paradigm in reinforcement learning, distinguished by their ability to overcome the limitations imposed by strictly on-policy rollouts. This architectural evolution builds on earlier

---

[10]More detailed results about Qwen3-4B and Qwen3-8B are in A.3.3

[11]AMD HIP is a CUDA-like C++ programming model that enables the same GPU codebase to run efficiently on both AMD (via ROCm) and NVIDIA GPUs.

[12]HIP-based implementation `torch_memory_saver`: https://github.com/fzyzcjy/torch_memory_saver

[13]`torch_memory_saver` document: https://github.com/sgl-project/sglang/issues/7009

advancements in classical methods, notably Asynchronous Advantage Actor-Critic (A3C) (Mnih et al., 2016) and IMPALA (Espeholt et al., 2018), which pioneered similar disaggregated designs. Recent frameworks such as AReaL (Fu et al., 2025), AsyncFlow (Han et al., 2025), StreamRL (Zhong et al., 2025a), and LlamaRL (Wu et al., 2025) have successfully adopted this asynchronous paradigm. More specifically, in an asynchronous RL framework, workers continuously stream rollouts generated by the inference engine into a shared buffer, while the training engine independently retrieves rollouts from this buffer for model updates. The disaggregation of inference and training engines across distinct groups of computational resources enables fine-grained allocation tailored to task-specific demands. This decoupling maximizes hardware utilization and has been shown to deliver substantial throughput improvements. However, it also introduces the challenge of rollout staleness, whereby an increased proportion of off-policy rollouts may be consumed. Such staleness may lead to training instability and, in some cases, to degraded convergence and reduced final accuracy in RL training.

The choice between synchronous and asynchronous paradigms presents a fundamental trade-off between training accuracy and performance. Inspired by this system-level design philosophy, our goal is to preserve the on-policy nature of RL required by many algorithms while simultaneously improving efficiency. To this end, we introduce APRIL, which *incorporates an asynchronous mechanism into the rollout generation process to mitigate the long-tail problem* and the associated issue of resource underutilization. Crucially, APRIL mitigates long-tail inefficiency without the complexities of full asynchronous decoupling or the burden of managing off-policy staleness. APRIL makes synchronous RL more efficient and robust, narrowing the gap with the performance of asynchronous approaches while preserving the benefits of on-policy rollouts. Our experiments further show that APRIL achieves comparable, and in some cases slightly higher, accuracy than the original synchronous framework.

**Rollouts: On-policy vs. Off-policy**   The algorithmic foundation of RL is intrinsically linked to the dichotomy of on-policy versus off-policy learning. This fundamental distinction determines how rollout is collected and subsequently used for policy model updates, thereby establishing a critical trade-off between training stability and accuracy versus sample-collection (rollout data) efficiency.

In on-policy learning, PPO (Schulman et al., 2017; Zheng et al., 2023) remains the standard for RL training of LLMs, but it requires fresh rollouts for each update, resulting in sample inefficiency and high computational cost. PPO-based training also involves four distinct LLMs—the policy, reference, reward, and value models—leading to substantial GPU memory overhead. To address these issues, GRPO (Shao et al., 2024; Mroueh et al., 2025) removes the explicit value network by averaging multiple reward-scored responses as a baseline, thereby reducing resource demands. Building on this, DAPO (Yu et al., 2025b) and GSPO (Zheng et al., 2025) extend GRPO to overcome its limitations.

In contrast, off-policy learning updates the policy using rollouts from previous policy versions, alleviating the rollout collection bottleneck but risking instability due to a distribution mismatch between the behavior and target policies. This mismatch can increase gradient variance and harm training stability. Nonetheless, recent work has explored hybrid approaches showing that, with proper management, off-policy data can preserve stability while improving performance. For example, LUFFY (Yan et al., 2025) augments on-policy RL with off-policy reasoning traces, balancing imitation and exploration while applying regularized importance sampling to sustain exploration. *Our method - APRIL's partial rollout strategy can be viewed as a mild relaxation of strict on-policy training*, gaining partial off-policy efficiency benefits while retaining the stability of on-policy RL.

## 5 CONCLUSION

In this work, we introduced APRIL, a simple yet effective approach to alleviating long-tail inefficiencies in reinforcement learning for large language models. By reusing partial rollouts and reducing wasted computation, APRIL improves throughput by **22.5**% on average across commonly used RL algorithms, accelerates convergence, and, in some cases, achieves an average of **2.1**% higher final accuracy across three tasks. In addition, we have made further engineering efforts to support deployment across diverse hardware devices. Beyond these empirical gains, APRIL highlights the importance of addressing both system-level bottlenecks and algorithmic design, offering a practical step toward more efficient and sustainable RL training. We anticipate that its principles will inspire more future research on scalable algorithms and system-level strategies for large-scale RL training.

## ETHICS STATEMENT

This work focuses on improving the efficiency of reinforcement learning (RL) for large-scale language models by mitigating long-tail inefficiencies in rollout generation. All datasets used in our experiments are publicly available and widely adopted in prior research. Our method, APRIL, is framework- and hardware-agnostic, and is intended to reduce computational overhead, thereby lowering energy consumption and improving sustainability in large-scale RL training. Potential ethical considerations include the broader risks associated with training powerful language models, such as misuse for generating harmful content. While our contribution primarily addresses system-level efficiency, we emphasize that responsible use of LLMs remains critical, and we encourage deployment of our method in alignment with established guidelines for ethical AI research and application.

## REPRODUCIBILITY STATEMENT

We have included our codebase in the supplementary material along with the paper submission. Readers and reviewers can follow the README file to reproduce the results presented in the paper. We also provide the complete hyperparameter settings in the Appendix. Here, we summarize the datasets, algorithms, and hyperparameters for easier review and replication.

Table 3: RL training hyperparameters. The left table shows the inference hyperparameters, while the right table shows the training hyperparameters.

(a) Standard (non-partial rollout) and APRIL (partial rollout) configuration

| Hyperparameter | Value |
|---|---|
| *Rollout Configuration* | |
| rollout_batch_size | 32 |
| n_samples_per_prompt | 8 |
| rollout_max_response_len | 16384 |
| rollout_temperature | 0.8 |
| n_samples_per_eval_prompt | 16 |
| eval_max_response_len | 16384 |
| eval_top_p | 0.7 |
| *Partial Rollout - specific hypermeter* | |
| over_sampling_batch_size | 64 |

(b) Training Configuration

| Hyperparameter | Value |
|---|---|
| *Training Configuration* | |
| global_batch_size | 256 |
| optimizer | Adam |
| lr | 1e-6 |
| weight_decay | 0.1 |
| adam_beta1 | 0.9 |
| adam_beta2 | 0.98 |
| kl_coef | 0.0 |
| eps_clip | 0.2 |
| eps_clip_high | 0.28 |

**Algorithms, Model, and Datasets** Our experiments are conducted on two models **Qwen3-4B** and[14] **Qwen3-8B** [15] with three math reasoning tasks. We use a diverse set of mathematical reasoning datasets for training, including **DAPO-Math-17k**[16], **DeepMath-103K**[17], and **DeepScaleR**[18]. To evaluate the final performance, we use the **AIME-2024** benchmark[19], a collection of recent challenging math reasoning problems. This benchmark serves as a standard for assessing the advanced mathematical reasoning abilities of large language models. To faithfully validate the robustness of the proposed method, we apply APRIL to the two most widely used RL algorithms: GRPO (Mroueh et al., 2025) and DAPO (Yu et al., 2025a).

**Hardware Platform** APRIL can operate effectively across diverse hardware platforms. We primarily develop and evaluate it on a single node equipped with either 8× NVIDIA H100 GPUs or 8× AMD MI300 GPUs. Due to page limitations, the experimental results presented in the following sections focus mainly on the AMD 8× MI300 configuration.

---

[14] https://huggingface.co/Qwen/Qwen3-4B

[15] https://huggingface.co/Qwen/Qwen3-8B

[16] https://huggingface.co/datasets/open-r1/DAPO-Math-17k-Processed

[17] https://huggingface.co/datasets/zwhe99/DeepMath-103K

[18] https://huggingface.co/datasets/agentica-org/DeepScaleR-Preview-Dataset

[19] http://huggingface.co/datasets/HuggingFaceH4/aime_2024

**Hyperparameter Setting**   As shown in Table 3, we summarize the hyperparameters used in our experiments. In our setup, `rollout_batch_size=32` specifies the number of input instances, which corresponds to the batch size for a single training step. With `n_samples_per_prompt=8`, each input instance generates multiple rollouts, resulting in a total of $32 \times 8 = 256$ samples collected per step. For APRIL, as mentioned earlier, we over-provision rollout requests by setting `over_sampling_batch_size=64` ($2\times$`rollout_batch_size`), meaning that 512 samples are requested from the inference engine, but the rollout process terminates once the first 256 rollouts are completed.

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

# A  APPENDIX

## A.1  USE OF LLMS

We used ChatGPT-5 solely as a general-purpose writing assistant to rephrase sentences and correct grammar and typographical errors in this paper. The model was not involved in research ideation, experimental design, data analysis, or drawing conclusions. The authors take full responsibility for the content of the paper.

## A.2  EXPERIMENTAL SETUP

Table 4: RL training hyperparameters. The left table shows the inference hyperparameters, while the right table shows the training hyperparameters.

(a) Standard (non-partial rollout) and APRIL (partial rollout) configuration

| Hyperparameter | Value |
|---|---|
| *Rollout Configuration* | |
| rollout_batch_size | 32 |
| n_samples_per_prompt | 8 |
| rollout_max_response_len | 16384 |
| rollout_temperature | 0.8 |
| n_samples_per_eval_prompt | 16 |
| eval_max_response_len | 16384 |
| eval_top_p | 0.7 |
| *Partial Rollout - specific hypermeter* | |
| over_sampling_batch_size | 64 |

(b) Training Configuration

| Hyperparameter | Value |
|---|---|
| *Training Configuration* | |
| global_batch_size | 256 |
| optimizer | Adam |
| lr | 1e-6 |
| weight_decay | 0.1 |
| adam_beta1 | 0.9 |
| adam_beta2 | 0.98 |
| kl_coef | 0.0 |
| eps_clip | 0.2 |
| eps_clip_high | 0.28 |

**Algorithms, Model, and Datasets**  Our experiments are conducted on two models **Qwen3-4B** and[20] **Qwen3-8B** [21] with three math reasoning tasks. We use a diverse set of mathematical reasoning datasets for training, including **DAPO-Math-17k**[22], **DeepMath-103K**[23], and **DeepScaleR**[24]. To evaluate the final performance, we use the **AIME-2024** benchmark[25], a collection of recent challenging math reasoning problems. This benchmark serves as a standard for assessing the advanced mathematical reasoning abilities of large language models. To faithfully validate the robustness of the proposed method, we apply APRIL to the two most widely used RL algorithms: GRPO (Mroueh et al., 2025) and DAPO (Yu et al., 2025a).

**Hardware Platform**  APRIL can operate effectively across diverse hardware platforms. We primarily develop and evaluate it on a single node equipped with either 8× NVIDIA H100 GPUs or 8× AMD MI300 GPUs. Due to page limitations, the experimental results presented in the following sections focus mainly on the AMD 8× MI300 configuration.

**Hyperparameter Setting**  As shown in Table 4, we summarize the hyperparameters used in our experiments. In our setup, rollout_batch_size=32 specifies the number of input instances, which corresponds to the batch size for a single training step. With n_samples_per_prompt=8, each input instance generates multiple rollouts, resulting in a total of $32 \times 8 = 256$ samples collected per step. For APRIL, as mentioned earlier, we over-provision rollout requests by setting over_sampling_batch_size=64 (2×rollout_batch_size), meaning that 512 samples are requested from the inference engine, but the rollout process terminates once the first 256 rollouts are completed.

---

[20]https://huggingface.co/Qwen/Qwen3-4B
[21]https://huggingface.co/Qwen/Qwen3-8B
[22]https://huggingface.co/datasets/open-r1/DAPO-Math-17k-Processed
[23]https://huggingface.co/datasets/zwhe99/DeepMath-103K
[24]https://huggingface.co/datasets/agentica-org/DeepScaleR-Preview-Dataset
[25]http://huggingface.co/datasets/HuggingFaceH4/aime_2024

## A.3 EXPERIMENTS

### A.3.1 PERFORMANCE - TOKEN THROUGHPUT

A central claim of our work is that APRIL can substantially accelerate the rollout phase of the RL.

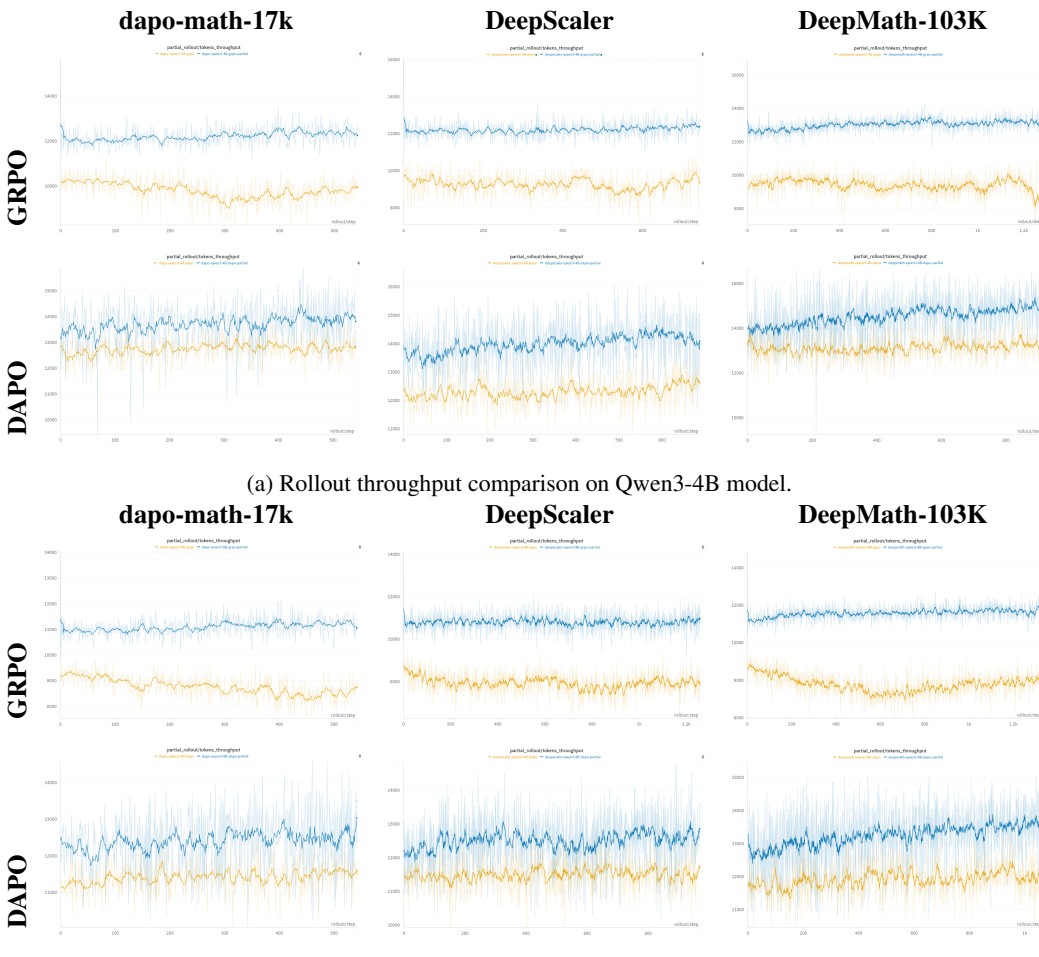

(a) Rollout throughput comparison on Qwen3-4B model.

(b) Rollout throughput comparison on Qwen3-8B model.

Figure 6: Comparison of rollout throughput between the baseline (non-partial rollout ——) and APRIL (——). The x-axis denotes the training steps, while the y-axis denotes the throughput. APRIL has the higher throughput across datasets: dapo-math-17k , DeepScaler, and DeepMath-103K.

We present a comprehensive evaluation of our active partial rollout (APRIL) represented with the line (——) against a standard non-partial rollout baseline represented with the line (——). We measure the rollout throughput, defined as the total number of tokens generated divided by the wall-clock time per rollout iteration. As shown in Figure 6, we list the actual throughputs. For the Qwen3-4B model, APRIL consistently improves throughput by **24.4**%, **31.8**%, **37.7**% with GRPO algorithm, and **9.0**%, **13.5**%, **9.8**% with DAPO algorithm across three datasets. As for Qwen3-8B model, the throughputs improve by **26.4**%, **34.7**%, **49.5**% and **8.7**%, **8.5**%, **10.2**%, respectively. Overall average throughput improves by around **22.5**%.

### A.3.2 CONVERGENCE SPEED AND ACCURACY

A potential concern with APRIL is that introducing off-policy rollouts (i.e., rollouts generated by earlier versions of the policy model, as described in Section 2.2 could destabilize training or degrade convergence and final accuracy.

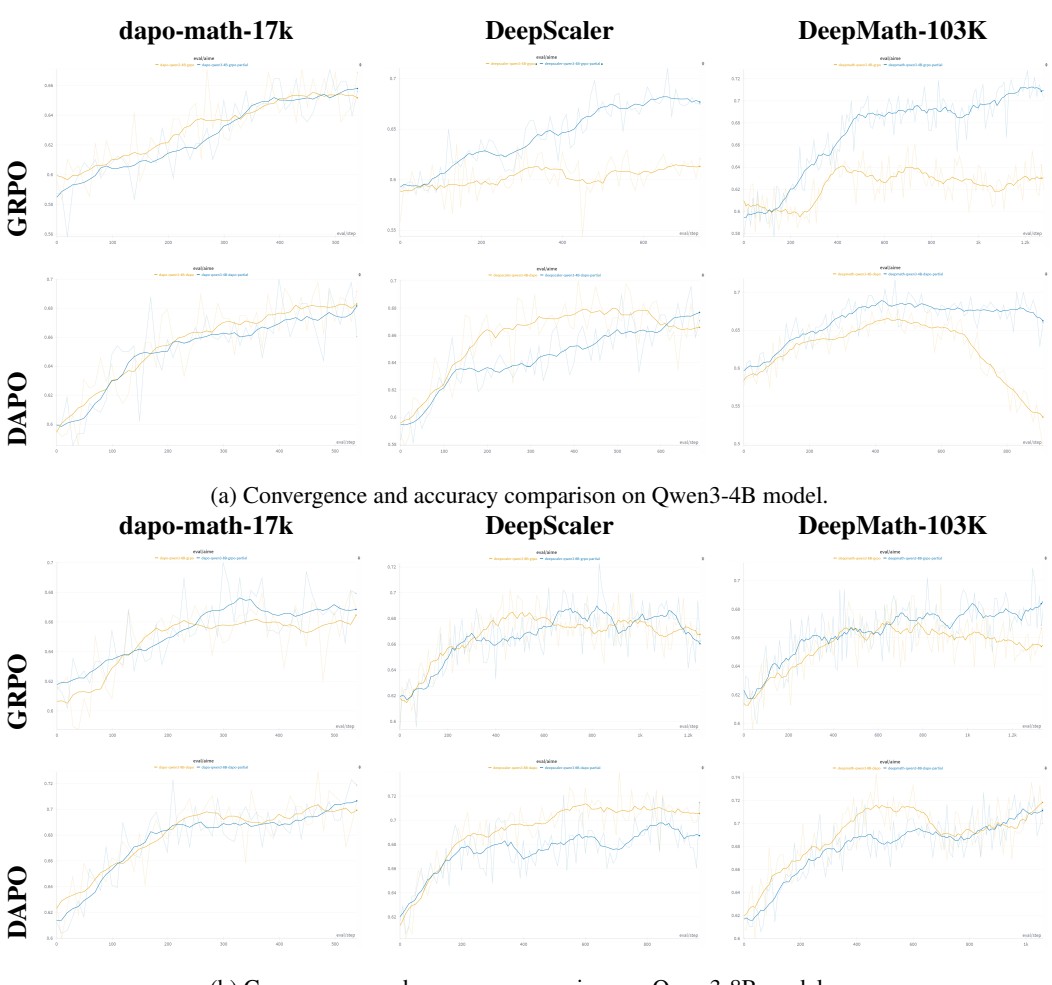

(a) Convergence and accuracy comparison on Qwen3-4B model.

(b) Convergence and accuracy comparison on Qwen3-8B model.

Figure 7: Comparison of convergence speed and accuracy between the baseline (non-partial rollout ——) and APRIL (——). The y-axis denotes the acc. Overall, APRIL performs better.

As illustrated in Figure 7, for the Qwen3-4B model, APRIL improves accuracy by **0.6**%, **1.1**%, and **8.2**% on GRPO, and by **-0.1**%, **1.2**%, and **12.8**% on DAPO. For the Qwen3-8B model, APRIL achieves gains of **0.4**%, **-0.7**%, **2.6**% on GRPO, and **0.7**%, **-1.6**%, **-0.4**% on DAPO. Overall average accuracy improves by around **2.1**%. This suggests that incorporating mildly off-policy rollouts enhances rollout diversity, thereby positively influencing both learning dynamics and final model performance. Moreover, training remains stable and can even benefit from the added diversity when updating policies within the RL framework.

### A.3.3  ANALYSIS OF PERCENTAGE OF PARTIAL ROLLOUTS AND ROLLOUT LENGTH

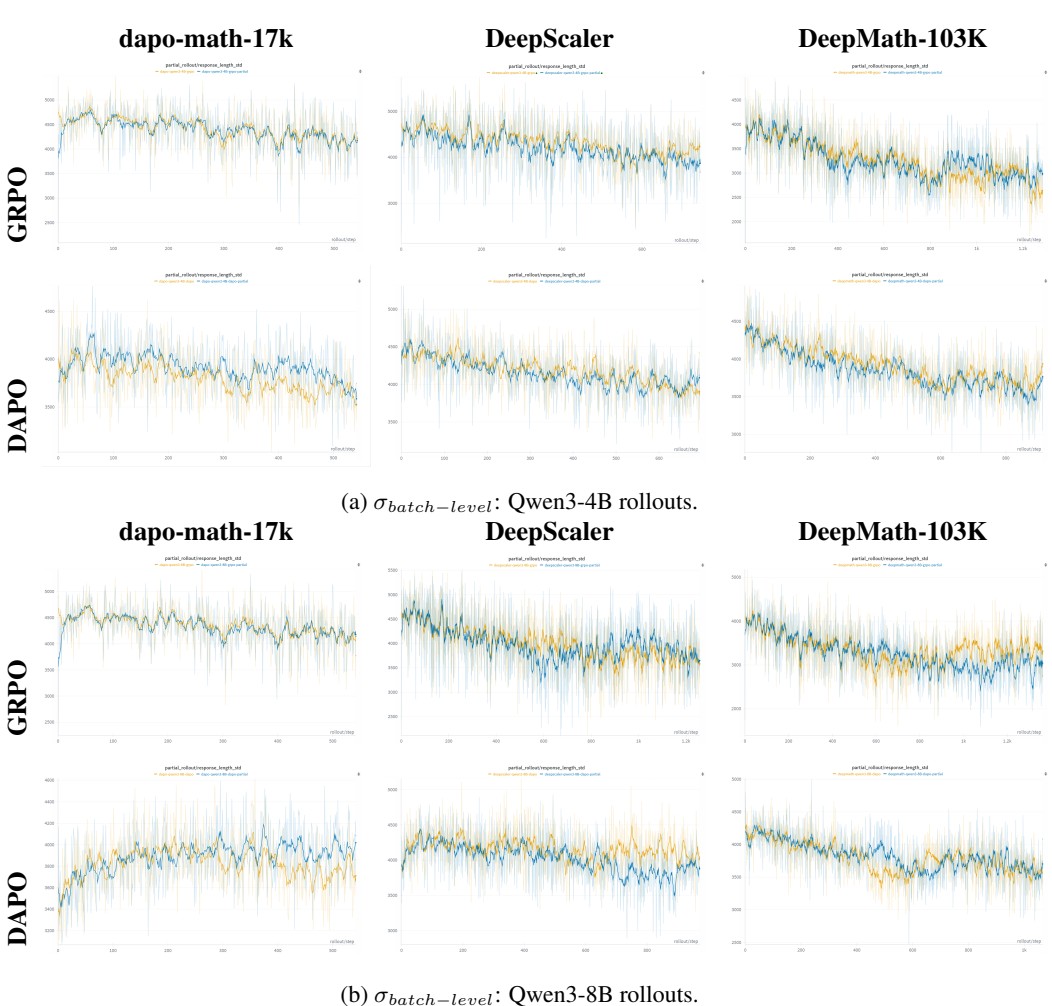

(a) $\sigma_{batch-level}$: Qwen3-4B rollouts.

(b) $\sigma_{batch-level}$: Qwen3-8B rollouts.

Figure 8: $\sigma_{batch-level}$: standard deviation of response length per iteration at the batch level.

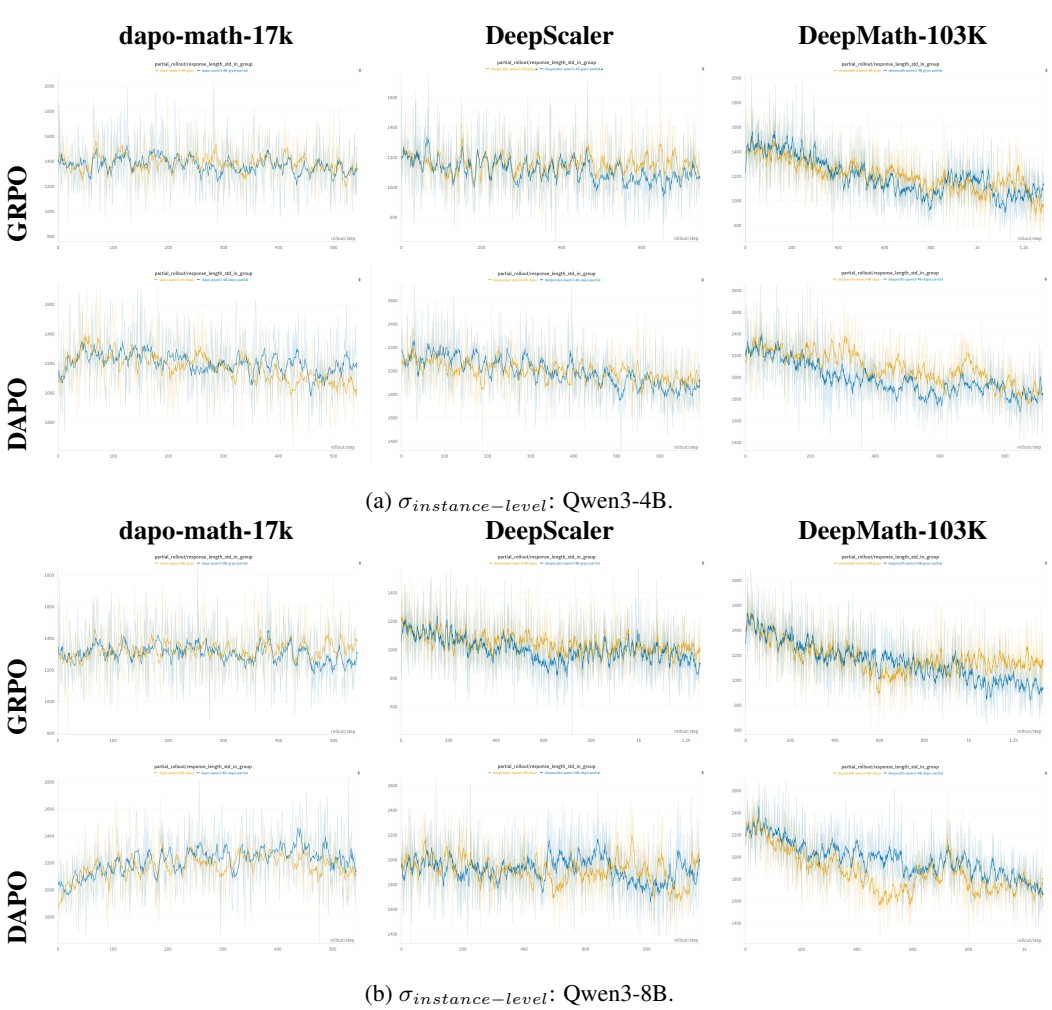

(a) $\sigma_{instance-level}$: Qwen3-4B.

(b) $\sigma_{instance-level}$: Qwen3-8B.

Figure 9: $\sigma_{instance-level}$: standard deviation of response length per iteration at the instance level.

### A.4 MOTIVATION AND MORE BACKGROUND

In the main paper, due to page limitations, we provided only a brief discussion to demonstrate the motivation. Here, we present a more comprehensive discussion by offering additional background and analyzing both system-level and algorithmic bottlenecks in existing RL frameworks. Specifically, we conduct a literature survey on RL framework architectures, existing rollout optimization techniques, and rollout categories to further clarify the motivation behind the design of APRIL.

#### A.4.1 RL FRAMEWORK ARCHITECTURES: SYNCHRONOUS VS. ASYNCHRONOUS

From an architectural perspective, RL frameworks can be broadly categorized into two types: synchronous and asynchronous. The primary distinction lies in the management of coordination and the allocation of computational resources between the inference engine (responsible for rollout generation) and the training engine (responsible for model weight updates).

**Synchronous RL** The mainstream paradigm in reinforcement learning (RL) systems is synchronous RL, as exemplified by frameworks such as TRL (von Werra et al., 2020), OpenRLHF (Hu et al., 2025), verl (Sheng et al., 2025), and ROLL (Wang et al., 2025). In this paradigm, the inference and training engines are typically co-located on the same GPUs, and the workflow is organized into sequential processes. First, when RL training is launched, a batch of instances is sent to the inference engine to generate rollouts. All workers simultaneously generate a full batch of experiences using a unified frozen policy model. Only after the entire batch has been collected does the policy update occur in the training engine. This design leverages the inherent on-policy nature of reinforcement learning, ensuring that the training process consistently utilizes the most recent rollout data. These rollouts, directly sampled from the current policy model, guarantee training stability and ultimately facilitate convergence. However, this simplicity comes at a considerable cost to efficiency. Specifically, the system must wait for all instances within a batch to complete rollout generation. Due to the long-tail phenomenon 3.2 in rollout lengths, some instances take substantially longer to finish, leading to idle time ("bubble") and severe underutilization of computational resources.

**Asynchronous RL** In response to the efficiency challenges inherent in synchronous RL, researchers have increasingly explored more dynamic and decoupled architectures. Fully asynchronous RL frameworks are emerging as a new paradigm in reinforcement learning, distinguished by their ability to eliminate the limitations imposed by strictly using on-policy rollouts in RL training. This architectural evolution draws heavily from earlier advancements in classical, notably Asynchronous Advantage Actor-Critic (A3C) (Mnih et al., 2016) and IMPALA (Espeholt et al., 2018), which pioneered similar disaggregated designs. Recent frameworks such as AReaL (Fu et al., 2025), AsyncFlow (Han et al., 2025), StreamRL (Zhong et al., 2025a), and LlamaRL (Wu et al., 2025) have successfully adopted this asynchronous paradigm. More specifically, in the asynchronous RL framework, workers continuously stream rollouts generated by the inference engine into a shared buffer, while the training engine independently retrieves rollouts from the buffer for model updates. The disaggregation of the inference and training engines onto distinct groups of computational resources enables fine-grained allocation tailored to task-specific demands. This decoupling maximizes hardware utilization and has been shown to deliver substantial throughput improvements. However, this paradigm also introduces the challenge of rollout staleness, wherein an increased proportion of off-policy rollouts may be utilized. Such staleness can lead to training instability and, in some cases, degraded final accuracy.

**Trade-offs Between Synchronous and Asynchronous RL** The choice between synchronous and asynchronous paradigms presents a fundamental trade-off between training stability and hardware utilization. RL systems like slime (THUDM, 2025) have been developed to offer a flexible architecture that supports both synchronous and asynchronous training modes within a single framework. This allows practitioners to choose the suitable mode according to the task types. Synchronous methods, with their controllability of rollout purity, are often preferred for tasks with relatively uniform completion times and static environments, such as mathematical reasoning or structured question-answering. While asynchronous systems are particularly advantageous for complex, interactive environments, such as multi-step agentic tasks, where low-latency adaptation is crucial and execution times are extremely heterogeneous. But due to the highly off-policy nature of fully asynchronous frameworks, they currently remain largely experimental implementations with significant limitations on usage scenarios. This makes the continued importance of synchronous frameworks crucial.

Inspired by this system-level design philosophy, our work, APRIL, *introduces an asynchronous mechanism in the rollout generation process to mitigate the aforementioned long-tail problem and the associated issue of resource underutilization. Crucially, it achieves this without incurring the complexities of full asynchronous decoupling or the additional burden of managing off-policy rollout staleness. This strategic intervention makes synchronous RL more efficient and robust for tasks with moderately variable generation times, thereby narrowing the efficiency gap with asynchronous approaches while retaining the critical benefits of the purity of the rollout on the policy.*

### A.4.2 ROLLOUT OPTIMIZATION

**Inference-level Speed-up**  Beyond the high-level architectural choice discussed, significant research has focused on optimizing the rollout stage itself. The autoregressive, token-by-token generation process of LLMs is inherently sequential and memory-intensive, making it a primary computational bottleneck in the RL training workflow. In traditional inference systems employing static batching, all requests within a batch must complete generation before the batch can be cleared. This constraint leads to GPU idling, as shorter sequences finish early and remain idle while waiting for longer sequences to complete. Recently, some optimizations have been introduced, for instance, Yu et al. (2022) proposed *continuous batching* to address the inefficiency of the traditional static batching by processing requests at the granularity of a single iteration or step. As soon as a sequence within the batch completes its generation, its allocated resources are immediately freed, and a new request from the queue can be added to the batch without delay. This dynamic approach significantly improves GPU utilization. More recently, *speculative decoding* (Leviathan et al., 2023; Chen et al., 2023; Svirschevski et al., 2025; Chen et al., 2025) has emerged as a powerful optimization. This method employs a smaller, faster draft model to generate a sequence of candidate tokens. The larger, more computationally expensive target model then verifies these proposed tokens. It accepts a prefix of correct tokens and, upon encountering the first mismatch, resamples from that point onward. This process effectively reduces the number of expensive forward passes required from the large model, thereby accelerating the overall generation speed. While these methods are powerful, their fundamental scope solely lies at the inference engine. They are designed to speed up the execution of individual rollouts by optimizing how tokens are generated or how batches are processed within a single generation task. By contrast, our work APRIL operates at a different, higher level of abstraction: the RL system scheduling layer. It does not modify the batching or decoding kernels. Instead, it introduces a scheduling mechanism that manages the lifecycle of rollout generation across multiple RL iterations. *Therefore, APRIL's partial rollout mechanism is complementary to the aforementioned inference-level optimizations, implying that it can be combined with these techniques for even greater overall efficiency gains in RL training.*

**System-level Scheduling**  Beyond the fine-grained improvements at the inference level, other system-level optimizations aim to address broader inefficiencies in the overall RL workflow. As previously discussed in Section, fully asynchronous RL (Noukhovitch et al., 2025) represents the most typical and intuitive approach to optimize RL workflows at this level by completely decoupling computational components. Within the synchronous RL settings, recent works such as SortedRL (Zhang et al., 2025) and RLHFuse (Zhong et al., 2025b) have explored system-level optimizations to mitigate these inefficiencies via length-aware scheduling and pipeline fusion mechanisms. The term partial rollout was first mentioned in the technical report for Kimi k1.5 (Kimi et al., 2025), where it was described as a technique to improve training efficiency for long-context RL. Specifically, it was conceptualized as "sampling new trajectories by reusing a large chunk of previous trajectories, avoiding the cost to re-generate the new trajectories from scratch". This concept was presented within the broader context of long context scaling and improved policy optimization. However, to the best of our knowledge, the specific mechanism, implementation, and its impact on system throughput and model convergence have not been detailed publicly. *Our work implements a specific form of this concept as a preemptive scheduling mechanism designed for the iterative loop of RL training.*

### A.4.3 ALGORITHM: ON-POLICY VS. OFF-POLICY

The algorithmic foundation of RL is intrinsically linked to the dichotomy of on-policy versus off-policy learning. This fundamental choice dictates how rollout is collected and subsequently used for policy model updates, thereby establishing a critical trade-off between training stability and sample efficiency.

**On-policy Rollout** Proximal Policy Optimization (PPO) stands as a cornerstone in RL training for LLMs (Schulman et al., 2017; Zheng et al., 2023), and is fundamentally an on-policy algorithm, which requires fresh data collection in each policy update for single use, leading to significant sample inefficiency and prolonged rollout times. Furthermore, the standard PPO-based RL training architecture typically necessitates the instantiation and management of four distinct LLMs: the policy model, a reference model, a reward model, and a value model. This ensemble contributes to substantial GPU memory overhead and considerable computational costs. To mitigate these computational and memory bottlenecks, Group Relative Policy Optimization (GRPO) (Shao et al., 2024; Mroueh et al., 2025) was introduced, which eliminates the need for an explicit value network by generating multiple candidate responses for each input instance and using their average score, derived from a reward function, as a baseline. This significantly reduces the computational requirements for RL on-policy training, making it more accessible and resource-effective.

**Off-policy Rollout** Off-policy learning, in contrast, allows the policy to be updated using rollouts generated by previous-version policy models, which solves the rollouts collection dilemma. However, many believe that this would harm the stability of training, since training on stale rollouts introduces a distribution mismatch between the behavior policy that generated the rollouts and the current target policy, thus leading to high variance in gradient estimates and unstable training. Despite these challenges, recent research has actively explored hybrid (or non-strictly on-policy) approaches, which demonstrate that with proper management, off-policy data can not only maintain training stability but also yield substantial benefits in training performance. LUFFY (Yan et al., 2025) is one such framework that augments on-policy RL with off-policy reasoning traces from stronger models. It dynamically balances imitation with exploration, and further incorporates policy shaping via regularized importance sampling to prevent superficial imitation and encourage sustained exploration throughout training. This mixed-policy method successfully compensated for the short board of the model in specific tasks and increased the exploratory the model. *Our APRIL- its partial rollout strategy can be interpreted as a mild form of breaking the strict on-policy, bringing partial off-policy efficiency benefits while maintaining the core stability advantages of on-policy RL algorithms.*

