# OpenReview forum: "APRIL: Active Partial Rollouts in Reinforcement Learning to Tame Long-tail Generation"
_ICLR.cc/2026/Conference — Submitted to ICLR 2026_

### Official Review · Reviewer_N1gK · 2025-10-28

**Soundness:** 2
**Presentation:** 3
**Contribution:** 2
**Rating:** 4
**Confidence:** 4

**Summary:**

This paper proposes a framework called Active Partial Rollouts in Reinforcement Learning (APRIL), designed to accelerate training for critic-free, policy-gradient RL algorithms. While the framework demonstrates notable speedups, the experimental evaluation is limited: the authors test only small models, a mathematics-focused dataset, and a setting that is effectively near on-policy.

**Strengths:**

1. The paper is well organized and easy to follow; the figures effectively aid comprehension.
2. The reported acceleration is strong and is demonstrated across different GPUs and algorithms.
3. The background section is thorough, providing a comprehensive review of relevant prior work.

**Weaknesses:**

1. The partial-rollout design appears problematic. As the authors acknowledge, off-policy samples can introduce instability. In my view, APRIL’s observed stability stems from updating sampled trajectories jointly within a single batch, effectively avoiding the need for importance sampling. If the authors adopted the DAPO setting—where each RL step uses a batch of 4,096 samples that are updated separately over 16 iterations—the policy would drift substantially after each update, and APRIL would likely become unstable. Nevertheless, such off-policy configurations are often necessary for larger models due to I/O constraints.
2. The evaluation is restricted to relatively small models (<8B) and math-related tasks; scalability and generality remain unverified.
3. Although minor, figure text should be at least as large as the main text. Moreover, screenshots from W&B or TensorBoard are not suitable as formal figures in an academic paper.
4. While I understand the motivation to highlight practical impact (e.g., merges into well-known GitHub repositories), this disclosure compromises the anonymity required by ICLR and should be removed.

**Questions:**

1. Can the authors report results under highly off-policy regimes, matching the DAPO configuration?
2. Can the authors evaluate the method on larger models and additional domains (e.g., coding and reasoning)?
3. Do the authors employ importance sampling to correct for off-policy data? The current setup appears mathematically inconsistent without it. If importance sampling is applied, how does it affect stability and performance?

**Details Of Ethics Concerns:**

The provided GitHub repository is not anonymized and violates the anonymity policy.

---

> ### Author Response · Authors · 2025-12-04
> **Reply N1gK**
>
> Thank you very much for your suggestions.
>
> Q1:Can the authors report results under highly off-policy regimes, matching the DAPO configuration?
>
> A1: Yes, we added to the appendix already.
>
>
> Q2:Can the authors evaluate the method on larger models and additional domains (e.g., coding and reasoning)?
>
> A2: Honestly, we did not have sufficient resources to train at then. But recently, we have had more comprehensive results on larger 32B and 100B+ models in coding and tool-using scenarios - I can add to the appendix in the camera-ready version. Besides, Shanghai AI Laboratory also uses this machine to train its P1 (Physics Olympiads) ---> I know you might not be satisfied with this answer; however, just let you know its adoption if you ask this not for reviewing and really want to try and have concerns in this.
>
>
> Q3:Do the authors employ importance sampling to correct for off-policy data? The current setup appears mathematically inconsistent without it. If importance sampling is applied, how does it affect stability and performance?
>
> A3: This is a great question. We actually implemented importance sampling as well (you can refer to our codebase). However, during training we found that APRIL is already quite stable and that off-policy mismatch does not significantly affect its performance. Even when we train larger models, APRIL remains stable; therefore, we did not report the importance sampling results in the paper, though we can add them in the appendix. Some of these findings were not included in the current version. We agree that off-policy data can have theoretical side effects, but these are often mild in real engineering practice. For example, in our experiments, even without using importance sampling (IS) to mitigate off-policy effects, APRIL remains stable as long as we use relatively high training precision (e.g., ≥ FP8). Practitioners applying APRIL at large scale (e.g., >100B-parameter models / MoE models) have similarly not observed severe issues in practice. We understand this answer may not fully address all of your theoretical concerns, but we hope it is helpful if you are considering adopting APRIL in practice and are worried about off-policy stability.
>
>
> Ethics Concerns: Finally, could you please clearly point out which GitHub repository is not anonymized? (We did not provide any APRIL link in the paper.) If you find it, please kindly let me know and let me remove it.

---

### Official Review · Reviewer_KyU3 · 2025-10-30

**Soundness:** 2
**Presentation:** 3
**Contribution:** 1
**Rating:** 2
**Confidence:** 4

**Summary:**

The paper proposes APRIL, a scheduling mechanism that aims to strike a balance between synchronous and asynchronous, on-policy and off-policy reinforcement-learning training for LLMs. It halts over-provisioned rollouts and buffers partial completions for continuation in later steps. The goal is to mitigate long-tail response-length stalls that underutilise GPUs. Reported results show a 22.5% gain in rollout throughput and even slightly better performance, despite some data becoming stale due to the scheduling.

**Strengths:**

The paper addresses an important bottleneck in post-training LLMs with RL. The authors aim to strike a balance between highly asynchronous scheduling—which increases data staleness—and fully synchronous RL approaches such as GRPO, which can be slow. The paper’s structure and exposition are clear, and the mechanism is well presented, though a few sections could be clarified further.

**Weaknesses:**

I have the following major concerns:
1. The “partial rollout” solution is not new. Both the Kimi K1.5 technical report and SortedRL have a similar architectural design: early termination of long-tail completions and buffering for continuation. Although the authors mention the two papers in the related work, it is descriptive but non-comparative.
2. Off-policy staleness is only empirically waved away. The throughput, compared to other async RL scheduling methods like AREAL and AsyncFlow, underperforms. So the soundness of this mechanism depends on its robustness to stale data with standard on-policy RL. For the empirical results, I can’t find explicit reports of multiple random seeds, CIs, or other statistical tests to verify empirical robustness. I think adding those experiments will strengthen the claim. But it still does not allay my concern regarding the unaddressed stale data, as the paper “No Representation, No Trust: Connecting Representation, Collapse, and Trust Issues in PPO” has stated that stale data can increase non-stationarity, ultimately leading to performance collapse. The phenomenon is also discussed in the recent paper “Prosperity before Collapse: How Far Can Off-Policy RL Reach with Stale Data on LLMs?”. The actionable item is to add some ablation experiments to explain to what degree staleness starts to hurt, and whether the model will collapse if trained longer.

Minor places:
1. At line 161, the phrase “rollout rollouts …” is confusing.
2. How the reward is handled for the incomplete rollouts is only implicitly mentioned at lines 88–89. My understanding is that the reward is not computed until the rollout is finished. I think this is worth stating more clearly.
3. In Figures 4–9, the plot labels are not readable unless you zoom in very closely.
4. The quotation marks are wrong, e.g., line 61 and line 1177.

**Questions:**

1. For the cached incomplete rollouts, how are the KV cache managed with the inference engine.
2. How is your method different from Kimi K1.5 and SortedRL?
3. Have you tried to train APRIL for longer, any policy collapsing observed?

---

> ### Author Response · Authors · 2025-12-04
> **Reply KyU3**
>
> Thank you very much for your suggestions.
>
> Q1: For the cached incomplete rollouts, how are the KV cache managed with the inference engine.
> A1: The KV cache from the last iteration would normally be discarded and recomputed using the new policy in the current iteration, which means partial rollouts do not cause any difference in KV cache management.
>
> Q2: How is your method different from Kimi K1.5 and SortedRL?
> We acknowledge that Kimi was the first to propose the concept of partial rollout, and our idea is inspired by their technical report (and they did not release the code and disclose implementation detail). However, our implementation is different. They break long responses into segments, which incurs much higher scheduling overhead, whereas our method can be seen as a more native and fine-grained improvement at the token level rather than the segment level. Moreover, they do not provide implementation details or open-source code. ---> I know you might not be satisfied with this answer; however, just let you know its adoption if you ask this not for reviewing and really want to try and have concerns in this.
>
> Q3: Have you tried to train APRIL for longer, any policy collapsing observed?
> However, during training we found that APRIL is already quite stable and that off-policy mismatch does not significantly affect its performance. Even when we train larger models, APRIL remains stable; therefore, we did not report the importance sampling results in the paper, though we can add them in the appendix. Some of these findings were not included in the current version. We agree that off-policy data can have theoretical side effects, but these are often mild in real engineering practice. For example, in our experiments, even without using importance sampling (IS) to mitigate off-policy effects, APRIL remains stable as long as we use relatively high training precision (e.g., ≥ FP8). Practitioners applying APRIL at large scale (e.g., >100B-parameter models / MoE models) have similarly not observed severe issues in practice. We understand this answer may not fully address all of your theoretical concerns, but we hope it is helpful if you are considering adopting APRIL in practice and are worried about off-policy stability.
>
> Finally, I would like to clarify one point: our goal is to mitigate the rollout bottleneck in RL training from a systems-design perspective, and this inevitably introduces some off-policy effects. We can use IS to mitigate this, and we have already implemented it in our codebase. However, these issues did not occur seriously in our experiments, so we did not discuss IS-based mitigation in detail in the paper due to space limitations. We will add these details to the appendix.

---

### Official Review · Reviewer_963w · 2025-11-09

**Soundness:** 4
**Presentation:** 3
**Contribution:** 3
**Rating:** 6
**Confidence:** 3

**Summary:**

The paper introduces APRIL, a novel off-policy method for RL training, designed to mitigate the inefficiencies caused by lengthy rollouts. APRIL employs over-provisions on rollout requests, initiates the training phase when a training batch is ready, and allows incompleted rollouts to resume in the later RL training steps. APRIL provides comprehensive evaluations to validate the effectiveness of its design.

**Strengths:**

1. The optimization techniques are well-reasoned, effectively addressing the important and timely problem of on-policy RL training, which may be inefficient when the data distribution is long-tailed.
2. The evaluation results of this paper are comprehensive, covering both the system side and algorithm side evaluations. The evaluation results demonstrate significant improvement in system throughput or algorithmic convergence over the chosen baseline.
3. The writing of this paper is easy to follow as well as rigorous.
4. The implementation of this paper is well-optimized; its open-sourced code will significantly contribute to the machine learning community.

**Weaknesses:**

1. The evaluation section only picks one baseline, which is strictly on-policy RL training. However, there are already some well-known frameworks (e.g., AReaL) that potentially employ a similar idea to this paper. The additional performance gains over these state-of-the-art frameworks are not explored.
2. Some of the experimental results might be contradictory to the previous approaches. For example, this paper claims that, with their off-policy method, the algorithm accuracy on the math reasoning task is superior to the strict on-policy method. While in the previous paper, e.g., AReaL, shows that without a specific algorithmic design (e.g., on the loss terms), the RL training performance can degrade significantly when incorporating stale data (i.e., rollouts that are generated with older versions of model parameters).
3. The optimizations introduced in this paper rely on an assumption that the rollout generation phase dominates the RL training. However, with larger inference batch sizes and more advanced GPUs (i.e., GPUs with larger GPU memory and GPU HBM memory bandwidth), will the rollout phase still dominate the RL training? For example, for an 8B model, one NVIDIA H200 GPU can accommodate a large batch size; with the strong HBM memory bandwidth capacity (4.8 TB/s on H200 GPUs), H200 GPUs can process rollouts very rapidly.

**Questions:**

Q1: What is the system throughput, as well as the algorithm convergence comparison of APRIL and previous state-of-the-art off-policy frameworks? If available, could the authors provide at least one additional baseline on one model?

Q2: Why is the algorithm convergence of APRIL superior to the on-policy method?  Some of the previous work claims that only with a careful algorithmic design could the off-policy method achieve a similar algorithm convergence. For example, in the AReaL paper, we can see that with more stale data, the algorithm's convergence drops significantly. The stale data, which may behave like a regularizer as stated by the APRIL paper, seems purely a drag down on algorithm convergence in AReaL. Even with AReaL's careful design (Sec 5 of their paper), the algorithm's convergence still cannot surpass synchronous RL training. As admitted by the APRIL authors, on average, there are 40% of stale data during training, while it's confusing that without a careful algorithm design, the algorithmic convergence can improve. Could the authors provide more insights?

Q3: What is the key difference between APRIL and AReaL? As stated in the APRIL paper, the maximum stale data used model parameters within 5 steps. What if AReaL adopts a small data staleness degree, e.g., 2 or 4 (Sec 5 of their paper). The two systems are operating similarly. Could the authors provide more analysis?

Q4: What is the percentage of time consumed on rollout generation when using more advanced hardware (and tuning to a new optimal inference batch size)? Does the APRIL's assumption still hold on NVIDIA H200 GPUs (if available, NVIDIA B200 GPUs or AMD MI350 GPUs)? This paper would benefit from more theoretical or empirical analysis of their assumption on more advanced GPUs.

---

> ### Author Response · Authors · 2025-12-04
> **Reply Reviewer 963w**
>
> Thank you very much for your suggestions.
>
> Reply:Q1: Actually, as we stated in the abstract and introduction, we are the first to implement this kind of method in the open-source community. At that time, we could not find another directly comparable baseline, but we did identify a related concurrent work (SortedRL) that was also submitted to ICLR. We will include a comparison with this work in the camera-ready version.
>
> Reply:Q2&Q3: AReaL is fully asynchronous, decoupling generation and training entirely, with continuous rollout production. This introduces significant staleness (e.g., a sample can be generated by a policy from many steps ago) and requires custom loss designs to mitigate collapse. Also, they use the log-probs generated by the inference engine for importance sampling, which can introduce another “off-policy” effect since the training and inference engines may use different numerical precision. APRIL remains synchronous overall but adds partial resumption as a mild off-policy relaxation (a sample is partially generated by the policy from the previous version). APRIL’s superior convergence compared to AReaL is due to this much smaller degree of staleness. Furthermore, the main problem is they did not release the code and did not disclose implementation details in the paper. Thus, we still have one more contribution is openness.
>
> Reply:Q4: Yes, it will still hold. It’s an interesting question. However, once you have worked with RL or run RL before, you probably know that rollout/inference is usually the main bottleneck and takes about 70–90% of the total time (in some cases, such as coding or agentic tasks, it can be even higher—over 90%). Thus, even if we adopt different hardware, the rollout bottleneck will still exist. On different hardware, we might instead focus on system-level aspects such as communication, bandwidth, or kernel optimizations to improve performance.

---

### Official Review · Reviewer_QftM · 2025-11-11

**Soundness:** 2
**Presentation:** 2
**Contribution:** 2
**Rating:** 2
**Confidence:** 3

**Summary:**

This work presents an asynchronous RL training framework for language models. To explain, in classical RL framework, we sample multiple responses for each prompt in each batch, and wait until the responses have been fully generated; in this work, the author proposes to over provision prompts for each batch, and stop rolling out once the number of rollouts reaches a threshold, and the incomplete rollouts are stored in the data buffer and are resumed in later steps. The authors run experiments with the proposed framework using qwen-3b/8b with mathematical datasets, and show improved efficiency and performance.

**Strengths:**

- **Significance**: Improving training efficiency and leveraging partial rollouts are important directions that many practitioners care about. Also, over-sampling prompts/states and using only a subset of them for gradient update in a batch has been used a lot in prior works, but mostly focusing on creating a learning curriculum as far as I know. This paper provides a different perspective by using the technique to accumulate more valid/complete rollouts within a fixed time window.

**Weaknesses:**

- **Quality**: Overall, this work feels like a technical report/blog, rather than a rigorous research paper with sufficient depth.
    - **Lack of experiments and formal analysis**: Off-policy correction is an important topic in RL, however there are no convincing analysis in the paper discussing the potential issue from the proposed asynchronous framework, and the current experiments provided does not provide sufficient evidence. Would the performance gain remains under different hyperparameters and benchmarks? Also, it may not be true that "slightly off-policy rollouts may act as a regularizer, preventing the policy from diverging into pathologically long generation modes"; i suggest the authors to consider quantifying the diversity/divergence and present more evidence to readers. It would be helpful to include a *technical* writeup (instead of qualitative description) on how prior works analyzing and addressing the off-policy issues, and discuss if this work could provide any formal guarantee on the convergence of the algorithm.
    - **Presentation**: The paper looks not to be ready for publishing. For example, Figure 4 and 5 looks to be directly coming from weight & bias platform. Please revise them for better readability.

- **Originality**: The proposed method is very much similar to an existing work [[Zhang et al., 2025](https://openreview.net/forum?id=YoV9lIZ827)]. Although the authors have discussed briefly in the related work, it is hard to see if there is any fundamental difference between the two.

**Questions:**

1. Algorithm: Would you please provide a formal algorithm block in the paper? This could enhance the readability of the work. Example: [[Phuong and Hutter, 2024](https://arxiv.org/pdf/2207.09238)].
2. Method: Would you please be specific on how the advantage term are estimated? I assume that it is done after the full completion, but it would be nice to formally write out the equations.
3. Figures: In addition to Figure 3, would it be possible to also plot rewards (or accuracy) v.s. length, and their changing trend as the training progresses.
5. Experiments: It would be meaningful to evaluate the scalability of the proposed method. I doubt if the performance would still be on par if the authors train longer with many different hyperparam combination / evaluate on more challenging benchmarks.
6. Analysis: Could you please provide formal analysis about the off-policy effect on the convergence of the algorithm?
7. Could you discuss in detail that how your work differs from prior works like sortedRL? The claim that "however, to the best of our knowledge, the specific mechanism ... have not been detailed publicly. Our work implements a specific form of this concept ..." is not convincing and cannot justify if this work has any original methodological contribution.

---

> ### Author Response · Authors · 2025-12-04
> **Reply Reviewer QftM**
>
> **Reply Weaknesses**: Thank you very much for your suggestions. Let us clarify our focus and see where we can revise the paper more concretely. Our primary goal is a practical contribution: a method that is easy to understand, addresses a real problem, discloses sufficient implementation details, and provides usable tools and code for the community. Accuracy is not the main objective; we aim to improve efficiency without a meaningful drop in accuracy. Empirically, across many RL algorithms, datasets, and hyperparameter settings (including multiple random seeds), we do not observe a significant accuracy degradation, and in some cases accuracy even improves slightly. If APRIL only worked in a small subset of cases, we would not see the overall gains reported. If your main concern is theoretical rigor (e.g., stronger proofs or a more formal tool setting), we would greatly appreciate more specific guidance—such as what form of “diversity/divergence” you would like to see and what scale of hyperparameter variation you consider appropriate (do you have in mind hundreds vs. thousands of combinations?). If your main concern is readability, we are very happy to revise the relevant sections and provide a clearer, camera-ready version. Regarding the methods section, we chose a relatively high-level formal description to keep it accessible. Other reviewers did not report difficulties here, but we are glad to add more details and clarifications in response to your comments. Finally, we agree that off-policy data can have theoretical side effects, but these are relatively mild in real engineering practice. For example, in this work, we obtain similar results with or without importance sampling (IS) to correct log-probability mismatch, as long as we avoid very low-precision training (e.g., ≤ FP8). Even when applying APRIL at large scale (e.g., >100B-parameter models), we have not observed severe issues in practice. You can verify our IS implementation in the provided codebase; we will add the corresponding discussion back into the paper and include it in the camera-ready version.
>
> **Reply:Q1**. Sure, I will add this
>
> **Reply:Q2**. At each step, we collect the completed rollout data (once the number of rollouts reaches the training batch size), and we compute the advantages at that point. Please refer to the updated PDF for details.
>
> **Reply:Q3**. I apologize if this was not clear. Have you had a chance to look at Figure 3 and Section 3.2 of the paper? Figure 3 corresponds to “3.2 The Long-Tail Problem in Rollout” and only involves inference; no training is performed in this experiment.
>
> **Reply:Q4**. Yes, it does, and we also provide corresponding results in the appendix. In addition, you can refer to several practical cases in the slime and miles GitHub repositories, where APRIL has been applied without observing any severe accuracy degradation.
>
> **Reply:Q5**. As I mentioned above about the Weaknesses, I can add this part in the paper; However, the theoretical off-policy effects may not be very severe in our setting. As mentioned, APRIL is primarily designed to improve performance rather than to fully eliminate off-policy effects on accuracy. If you feel this should be addressed more explicitly, we can incorporate importance sampling (IS) into APRIL to further mitigate the off-policy bias. As you know from practical RL systems in industry, it is common to use asynchronous training and rely on importance sampling or similar mechanisms to handle off-policy data. In contrast, APRIL focuses on synchronous RL training while still benefiting from the rollout efficiency typically associated with asynchronous approaches.
>
> **Reply:Q6**. These two methods are similar in their overall goal. SortedRL uses length-aware scheduling to mitigate long-tail effects, together with a cache to control the degree of off-policy data. In contrast, our approach over-provisions rollouts, actively terminates incomplete trajectories once the target is met, and then resumes them in subsequent iterations—without discarding data or relying on length prediction. We provide a minimal mechanism (oversubscription + early termination + prioritized continuation) that can be easily integrated into existing frameworks without requiring a new RL infrastructure. Also, as you can see from the SortedRL paper, it has been submitted to ICLR in the same cycle, so it is concurrent and comparable work. We will add an appropriate citation and include a discussion of SortedRL in our revised version.
>
> Finally, I would like to clarify one point: our goal is to mitigate the rollout bottleneck in RL training from a systems-design perspective, and this inevitably introduces some off-policy effects. We can use IS to mitigate this, and we have implemented it in our codebase. However, these issues did not occur seriously in our experiments, and because of page limitations; thus, we discard them. we will add it back in the camera-ready version.

---

> ### Author Response · Authors · 2025-12-04
> **Reply QftM**
>
> **Reply Weaknesses**: Thank you very much for your suggestions. Let us clarify our focus and see where we can revise the paper more concretely. Our primary goal is a practical contribution: a method that is easy to understand, addresses a real problem, discloses sufficient implementation details, and provides usable tools and code for the community. Accuracy is not the main objective; we aim to improve efficiency without a meaningful drop in accuracy. Empirically, across many RL algorithms, datasets, and hyperparameter settings (including multiple random seeds), we do not observe a significant accuracy degradation, and in some cases accuracy even improves slightly. If APRIL only worked in a small subset of cases, we would not see the overall gains reported. If your main concern is theoretical rigor (e.g., stronger proofs or a more formal tool setting), we would greatly appreciate more specific guidance—such as what form of “diversity/divergence” you would like to see and what scale of hyperparameter variation you consider appropriate (do you have in mind hundreds vs. thousands of combinations?). If your main concern is readability, we are very happy to revise the relevant sections and provide a clearer, camera-ready version. Regarding the methods section, we chose a relatively high-level formal description to keep it accessible. Other reviewers did not report difficulties here, but we are glad to add more details and clarifications in response to your comments. Finally, we agree that off-policy data can have theoretical side effects, but these are relatively mild in real engineering practice. For example, in this work, we obtain similar results with or without importance sampling (IS) to correct log-probability mismatch, as long as we avoid very low-precision training (e.g., ≤ FP8). Even when applying APRIL at large scale (e.g., >100B-parameter models), we have not observed severe issues in practice. You can verify our IS implementation in the provided codebase; we will add the corresponding discussion back into the paper and include it in the camera-ready version.
>
> **Reply:Q1**. Sure, I will add this
>
> **Reply:Q2**. At each step, we collect the completed rollout data (once the number of rollouts reaches the training batch size), and we compute the advantages at that point. Please refer to the updated PDF for details.
>
> **Reply:Q3**. I apologize if this was not clear. Have you had a chance to look at Figure 3 and Section 3.2 of the paper? Figure 3 corresponds to “3.2 The Long-Tail Problem in Rollout” and only involves inference; no training is performed in this experiment.
>
> **Reply:Q4**. Yes, it does, and we also provide corresponding results in the appendix. In addition, you can refer to several practical cases in the slime and miles GitHub repositories, where APRIL has been applied without observing any severe accuracy degradation.
>
> **Reply:Q5**. As I mentioned above about the Weaknesses, I can add this part in the paper; However, the theoretical off-policy effects may not be very severe in our setting. As mentioned, APRIL is primarily designed to improve performance rather than to fully eliminate off-policy effects on accuracy. If you feel this should be addressed more explicitly, we can incorporate importance sampling (IS) into APRIL to further mitigate the off-policy bias. As you know from practical RL systems in industry, it is common to use asynchronous training and rely on importance sampling or similar mechanisms to handle off-policy data. In contrast, APRIL focuses on synchronous RL training while still benefiting from the rollout efficiency typically associated with asynchronous approaches.
>
> **Reply:Q6**. These two methods are similar in their overall goal. SortedRL uses length-aware scheduling to mitigate long-tail effects, together with a cache to control the degree of off-policy data. In contrast, our approach over-provisions rollouts, actively terminates incomplete trajectories once the target is met, and then resumes them in subsequent iterations—without discarding data or relying on length prediction. We provide a minimal mechanism (oversubscription + early termination + prioritized continuation) that can be easily integrated into existing frameworks without requiring a new RL infrastructure. Also, as you can see from the SortedRL paper, it has been submitted to ICLR in the same cycle, so it is concurrent and comparable work. We will add an appropriate citation and include a discussion of SortedRL in our revised version.
>
> Finally, I would like to clarify one point: our goal is to mitigate the rollout bottleneck in RL training from a systems-design perspective, and this inevitably introduces some off-policy effects. We can use IS to mitigate this, and we have implemented it in our codebase. However, these issues did not occur seriously in our experiments, and because of page limitations; thus, we discard them. we will add it back in the camera-ready version.

---

### Meta-Review · Area_Chair_U6Ex · 2026-01-07

**Summary:**

The reviewers raised four major concern clusters that shaped the decision:
* Lack of Theoretical Rigor & Formal Analysis (QftM, N1gK): The paper was criticized as a "technical report" lacking depth. Key demands included: formal algorithm block, explicit advantage estimation equations, quantified analysis of off-policy diversity/divergence, and theoretical convergence guarantees rather than qualitative statements.
* Off-Policy Stability & Staleness (All reviewers): While authors claimed mild staleness acts as a regularizer, reviewers flagged insufficient evidence. Concerns included: no importance sampling (IS) implementation details, lack of ablation on staleness degree, contradictory claims vs. prior work (AReaL) showing off-policy data hurts convergence, and insufficient statistical validation (multiple seeds, CIs).
* Originality & Prior Work Differentiation (QftM, KyU3, N1gK): Reviewers questioned novelty versus Kimi K1.5 (token-level vs. segment-level) and SortedRL (concurrent ICLR submission). The claim "first open-source implementation" was deemed insufficient methodological contribution. Direct experimental comparisons with these methods were missing.
* Limited Evaluation Scope (963w, KyU3, N1gK): Concerns included: single baseline (strict on-policy), no comparison with SOTA off-policy frameworks (AReaL, AsyncFlow), small-scale models only (≤8B), math-domain exclusivity, and unresolved scalability claims for advanced hardware (H200/B200). Figures were deemed unpolished (W&B screenshots, unreadable labels).

**Reviewer Concerns:**

Addressed:
* Differentiation with prior work: The authors provided conceptual distinctions from SortedRL (oversubscription+resumption vs. length-aware caching) and AReaL (synchronous vs. fully async), but lacking experimental head-to-head validation.
* IS Implementation: The authors acknowledged IS code exists and will be added to appendix, but did not demonstrate its necessity or impact.
* KV Cache Management: Clarified recomputation policy (KyU3 Q1).

Largely Unaddressed/Outstanding:
* Theoretical rigor: No concrete plan for formal convergence analysis or quantified diversity metrics; the "practical contribution" defense didn't satisfy reviewers seeking mathematical depth (QftM Q5, N1gK Q3).
* Comparative evaluation: No new baseline experiments (e.g., AReaL with low staleness) or hardware scalability analysis (H200) to substantiate throughput claims (963w Q1/Q4, KyU3 Q2).
* Comparative evaluation: No new baseline experiments (e.g., AReaL with low staleness) or hardware scalability analysis (H200) to substantiate throughput claims (963w Q1/Q4, KyU3 Q2).
* Scale & generality: Promised results on 32B/100B models for coding "can be added" but not provided; insufficient to counter domain/model-size limitations (N1gK Q2).

**Reviewer Scores:**

Reviewer QftM:
Would likely stay at 2 (no change). While the rebuttal promised a formal algorithm block and improved figures, the core demand for rigorous theoretical analysis and deeper off-policy discussion, was met with "practical contribution" framing rather than mathematical substance. The reviewer explicitly sought proofs and formal tools, not just cosmetic improvements.

Reviewer 963w:
Would likely stay at 6 (no change). The reviewer appreciated the practical value and comprehensive evaluation, but the lack of new experimental data comparing against AReaL (even with low staleness) or validating hardware scalability claims remains unaddressed. The conceptual explanation of differences was helpful, but without empirical backing, a confident raise above "marginal" is unlikely.
Reviewer KyU3:

Would likely stay at 2 (no change). The originality concern—partial rollouts being neither new nor sufficiently differentiated from Kimi K1.5/SortedRL—persisted despite the rebuttal's "token-level vs. segment-level" distinction. No comparative experiments were provided, and the empirical robustness claim ("stable in practice") lacked statistical validation (multiple seeds, CIs). The contribution score of 1 ("poor") would be hard to raise on promises alone.

Reviewer N1gK:
Would likely increase to 4 or 6. The concerns were more pragmatic: evaluation scope and off-policy stability under harsh conditions (DAPO config). The rebuttal's promise to include DAPO results and large-scale (32B/100B) experiments, if believed, could satisfy their push for broader validation. The acknowledgement of IS implementation, even if unused, also partially addresses mathematical consistency.

---

### Decision · Program_Chairs · 2026-01-26

Reject